# COVID19 Sero-epidemiology and vaccine uptake in two Southern Nigerian clinics

Chinedu A. Ugwu[1,2], Ooreofe Odebode[1,2], Israel O. Ajayi[1,2], Josiah Olubunmi[1,2], Bejide Ifeoluwa[1,2], Jolly A. Adole[1,2], Kazeem Akano[1,2], Johnson Okolie[1], Philomena Eromon[1], Imonikhe K. Kio[1], Judith Amadi[1], Precious F. Adebayo[1], Iheanyi Mgbeokwere[1], Akeem O. Lawal[3], Olayemisi A. Adegoke[3], Amoke O. Adeyemi[1], Sampson Owhin[4], Oladele O. Ayodeji[4], Alebiosu Ahmed[1], Etim Henshaw[1], Oyejide Nicholas Eyitayo[1], Abah A. Sylvester[1,4], Abiola Omidele[1,4], Kolade Emilola[1], Olayiwola Funmilayo[1], Adeboye Oyewale Rufus[1], Oluwatuyi O. Olakunle[1], Obaado Oluwaseun[1], Angela M. Matta[5], Prajakta Bedekar[6,7], Latifeh Dahmash[8], Catherine S. Forconi[5], Ann M. Moormann[5], Rhoel R. Dinglasan[8], Christian T. Happi[1,2], Raquel A. Binder[5]*

1 The African Centre of Excellence for Genomics of Infectious Diseases, Redeemer's University Nigeria, Ede, Nigeria, 2 Department of Biological Sciences, Redeemer's University, Ede, Nigeria, 3 Ikorodu General Hospital, Ikorodu, Nigeria, 4 Federal Medical Centre Owo, Owo, Nigeria, 5 Department of Medicine, University of Massachusetts Chan Medical School, Worcester, Massachusetts, United States of America, 6 Department of Applied Mathematics and Statistics, Johns Hopkins University, Baltimore, Maryland, United States of America, 7 Department of Mathematics, Birla Institute of Technology and Science Pilani, Hyderabad, Telangana, India, 8 Department of Infectious Diseases and Immunology, College of Veterinary Medicine, University of Florida, Gainesville, Florida, United States of America

* raquel.binder@umassmed.edu

## Abstract

Africa has reported lower COVID-19 morbidity and mortality compared to other regions. Yet, the flat COVID-19 trajectory remains to be fully explained. Nigeria, Africa's most populous country, accounted for less than 1% of global COVID-19 cases. The present study evaluated SARS-CoV-2 seroprevalence, vaccine uptake, and methodological differences in serological threshold setting among febrile patients attending peri-urban and urban clinics in Southern Nigeria. Between October 2022 and February 2023, 745 febrile participants were recruited from Owo (Ondo State) and Ikorodu (Lagos State). Serum samples were tested for IgG antibodies against wild-type and variant SARS-CoV-2 antigens using a multiplex bead-based antibody assay. Seropositivity was calculated using two classification boundary methods: (i) the conventional three standard deviations above the mean of negative controls (3STDV) and (ii) the open-source SeroNIST tool. Vaccine uptake and risk factors for exposure were assessed. Vaccine uptake was 66.0% and significantly higher in peri-urban Owo (80.2%) than urban Ikorodu (52.6%, p-value < 0.0001). Increasing age was positively associated with vaccination (p-value < 0.0001). Sex, education, and religion did not influence vaccine uptake. Vaccinated individuals had significantly lower odds of being positive for anti-N SARS-CoV-2 antibodies (p-value = 0.001), indicating a potential protective effect of the vaccine. Age, education, and religion did

**Data availability statement:** All relevant data are within the paper and its Supporting information files.

**Funding:** Funding was provided by (1) the Centers for Disease Control and Prevention (CDC) through the U01GH002338-01 grant (RRD, CTH, and AMM) titled "Seroepidemiological Insight into COVID-19 Transmission in Africa (SICA)", (2) the National Institute of Health (NIH) through the National Center for Advancing Translational Sciences KL2-TR001455 grant (RAB), and (3) the Bill and Melinda Gates Foundation through the GIISER Project ARISE-INV-036304 (CTH and CAU). The funders had no role in study design, data collection and analysis, decision to publish, or preparation of the manuscript.

**Competing interests:** The authors have declared that no competing interests exist.

not influence anti-N SARS-CoV-2 seropositivity. Overall, our study revealed high S/RBD-based seroprevalences for all variants except Omicron among the unvaccinated participants (range: 89.1% - 100%). SARS-CoV-2 exposure in Southern Nigeria was likely widespread and underrecognized, with high seroprevalence among symptomatic peri-urban and urban populations. Peri-urban communities demonstrated higher vaccine uptake than urban counterparts, and age-targeted campaigns likely drove uptake among older individuals. Further, our study detected a potential protective effect of the vaccine among the study population. Finally, standardized thresholding methods such as SeroNIST are critical for accurate seroprevalence estimates.

## Introduction

Severe acute respiratory syndrome coronavirus 2 (SARS-CoV-2), causing coronavirus disease 2019 (COVID-19), was first identified in Wuhan, China, in December of 2019 and has since rapidly spread across the globe, causing the biggest pandemic of the 21st century [1]. Close to 800 million cases and around 7 million deaths have been attributed to COVID-19, which is likely underestimating the true global burden [2]. Surprisingly, Africa experienced a less catastrophic outbreak compared to the rest of the world, accounting for less than 2% of the global cases [3]. Nigeria, the most populated country in Africa, accounted for just 0.003% of the global cases [2]. At first, limited testing, a younger population, reduced transmission, preexisting immunity, genetic protective factors, early implementation of public health measures, and timely leadership were proposed to drive the flat COVID-19 trajectory [4,5]. More recent evidence suggests that reduced viral transmission and poor surveillance were not the main reasons the reported COVID-19 morbidity and mortality were low [6]. Instead, the population age structure and a reduced anti-SARS-CoV-2 inflammatory response due to specific immunoregulation or cross-reactive pre-pandemic immunity are likely the main protective factors [6]. Similar factors were likely the reason a low number of COVID-19 cases and related deaths were reported in Nigeria [7,8]. To better understand the extent of existing immunity to SARS-CoV-2 in Nigeria and risk factors of infection, we implemented a study measuring seroprevalence against different variants of SARS-CoV-2 between October 2022 and February 2023 in two Southern Nigerian clinics covering peri-urban and urban settings. Given the varying vaccine availability and acceptability in the region, we also evaluated the COVID-19 vaccine uptake. At the time of the study, the following vaccines were approved for emergency use in Nigeria and available in our study population: Vaxveria (Oxford/ AstraZeneca), Covishield (Serum Institute of India), Comirnaty (Pfizer/BioNTech), Jcovden (Johnson and Johnson), Spikevax (Moderna), and Sputnik V (Gamaleya). Note, Covilo (Sinopharm) was approved but not used in our study population. By 2023, ~110 million doses of COVID-19 vaccines had been distributed and ~30% of the population had full vaccine coverage [9,10].

Sero-epidemiological studies (serosurveys) evaluate pathogen-specific antibodies, thus serving as a tool to evaluate the burden of infection and existing immunity

among populations. Serosurveys are particularly valuable in contexts where clinical reporting and molecular testing for infectious diseases are constrained [11]. There are limited SARS-CoV-2 serosurveys implemented in Nigeria, and most of them focused on healthcare workers or only assessed the ancestral SARS-CoV-2 strain (i.e., wildtype [WT] or Wuhan) [11–16]. Between February 2020 and November 2022, Nigeria had experienced four major COVID-19 waves, which were dominated by the wildtype (WT)/ancestral, Alpha, Delta, Eta, and Omicron variants [17–20]. Other important variants detected elsewhere, such as Gamma and Lambda, were not reported in Nigeria. Whether this means these variants did not circulate in Nigeria or whether they went unreported is unclear.

While serosurveys are essential public health tools, the applied downstream analysis methods to convert quantitative outcomes (e.g., median fluorescence Intensity [MFI] values from bead-based assays or optical density [OD] values from enzyme-linked immunosorbent assays [ELISAs]) into qualitative results (positive/negative) vary greatly, making met-analyses and comparisons of outcomes across studies and regions challenging [21,22]. Here, we applied a validated weighted-standard curve across-plate normalization method [23] and compared two classification boundary methods for optimal qualitative serological assessments: (1) The traditionally applied three standard deviations above the mean of the negative controls (3STDV), and (2) an open-source machine learning-based tool that calculates sensitive and specific classification boundaries based on population-specific positive and negative controls, which has been termed SeroNIST [22–25].

In summary, we implemented a 2022/2023 SARS-CoV-2 serosurvey in Southern Nigeria based on standardized sero-logical tools, determined the risk factors for infection, and evaluated the COVID-19 vaccine uptake among symptomatic individuals visiting local health care centers. Understanding the sero-epidemiological dynamics of the disease and con-trasting peri-urban and urban settings is crucial for assessing the population's exposure, immunity levels, and potential risk factors. By examining the SARS-CoV-2 seroprevalence in Southern Nigeria, this study bridged knowledge gaps about regional risk factors for infection and population-level exposures. Finally, assessing COVID-19 vaccine uptake allowed us to evaluate coverage and inform future public health strategies aiming to increase vaccine uptake.

## Methods

### Study clinics

The study was conducted at two clinical sites in Southern Nigeria: (1) The Federal Medical Centre Owo (FMC Owo), in Ondo State from October 19 to November 22, 2022, and (2) the Ikorodu General Hospital (Ikorodu GH) in Lagos State from November 14, 2022, to February 15, 2023. These sites were selected based on their capacity to support surveil-lance research among febrile populations and their existing clinical diagnostic and research infrastructure. Both facilities serve large catchment populations and represent diverse epidemiological settings, covering both peri-urban (Owo) and urban (Ikorodu) sites, thus offering a unique opportunity to assess SARS-CoV-2 exposure and COVID-19 vaccine uptake. Additional information regarding the ethical, cultural, and scientific considerations specific to inclusivity in global research is included in the Supporting Information, see the Supplemental Checklist (S1 Checklist).

FMC Owo is a federal tertiary healthcare institution with peri-urban extension services and a fully functional laboratory run by the African Centre of Excellence for Genomics of Infectious Diseases (ACEGID). The facility supports a wide range of diagnostic capabilities and biological sample processing. Ikorodu GH is a high-volume secondary health facility serving an urban population just outside metropolitan Lagos, with clinical services and infrastructure well-suited to the project's needs.

Both sites provided centralized coordination of participant enrolment, data collection, and sample processing. Field teams were trained in Good Clinical Practice (GCP), biosafety procedures, and the use of standardized data collection tools. Community engagements were carried out at both sites to ensure culturally appropriate and voluntary participation. The study design ensured contextual relevance, operational feasibility, and scientific rigor for characterizing the SARS-CoV-2 epidemiology and COVID-19 vaccine uptake in distinct healthcare settings within southern Nigeria.

## Human subjects

The study protocol, informed consent procedures, and all associated study tools were reviewed and approved by the Nigerian National Health Research Ethics Committee (NHREC) and the Health Research Ethics Committees (HRECs) of FMC Owo (NHREC/01/01/2007-31/08/2021). Similarly, the University of Florida Institutional Review Board (IRB) reviewed and approved the study and associated documents (IRB202102308). Ethical oversight covered all aspects of study implementation, including participant recruitment, data collection, and biological sampling.

Inclusion criteria: febrile individuals across all ages seeking care at either study clinic. All participants provided written informed consent prior to enrolment. For participants under the age of 18, assent was obtained in conjunction with parental or guardian consent. Trained field personnel conducted the consent process using IRB-approved forms. The consent process clearly outlined the study objectives, procedures, potential risks, and the voluntary nature of participation. Participants were informed of their right to withdraw from the study at any time without penalty. Confidentiality was maintained throughout the study. Each participant was assigned a unique identifier, which was used to label all data and specimens. Electronic data were entered into encrypted REDCap databases hosted on password-protected devices. To mitigate risks, field staff were trained in infection prevention and control practices and used personal protective equipment during all participant interactions. These measures ensured ethical standards were upheld while maximizing participant safety and study integrity.

## Sample and data collection

Participant recruitment occurred through the outpatient departments of both hospitals. Individuals presenting with acute febrile illness were screened for eligibility (i.e., consenting abilities) and invited to participate. Post-consent, a brief survey (capturing basic demographics and COVID-19 vaccine uptake) and a blood sample were collected. Each participant provided finger-prick whole blood through an EDTA microtainer tube. Serum was separated onsite using portable centrifuges and stored -20 °C under cold-chain conditions. Each sample was labelled with a unique study identifier and linked to participant data through encrypted REDCap databases. Stored and processed samples were transported from the site using a dry shipper (pre-charged to -20 °C) to the ACEGID laboratory at Redeemer's University for bead-based multiplex serology (Luminex MagPix).

### *Serology* - Multiplex bead-based antibody assay

A bead-based IgG SARS-CoV-2 multiplex assay was implemented as previously described [23]. Briefly, wild-type (WT; Wuhan) full-length spike (S), WT nucleocapsid (N), WT receptor-binding domain (RBD), RBD Alpha, RBD Beta, RBD Gamma, RBD Delta, RBD Lambda, and RBD Omicron SARS-CoV-2 antigens were coupled to MagPlex microspheres (Luminex Corporation). Bovine Serum Albumin (BSA) coupled beads were used to control for bead-specific background noise. All coupled beads were diluted in ABE buffer (PBS, BSA 0.1%, TWEEN 20%, Sodium Azide 0.05%) to make a master mix of all the coupled antigens (consistent concentration across all beads, 1,000,000 beads/mL). Then, the master mix was added to each well of a 96-well plate and placed on a magnet to allow the beads to settle and remove the ABE buffer. Subsequently, 50 µL of human serum samples diluted (1:100) in ABE buffer were added to the wells, followed by a 2-hour incubation period on a plate shaker (300 rpm). Next, the beads were washed three times with 125 µL ABE buffer while plate remained on the magnet. Then 50 µL of biotinylated anti-human secondary IgG (BD Pharmingen) diluted in ABE (1:1000) was added to each well and followed by a one-hour incubation at room temperature on a plate shaker (300 rpm). The beads were once again washed three times, and 50 µL of phycoerythrin-conjugated streptavidin (1:1000; BD Pharmingen catalog # 554061) diluted in ABE added to the wells. Next, the beads were set on the plate shaker at 300 rpm for 15 minutes. After the final incubation period, the beads were washed 3 times for the last time and resuspended in 125 µL of ABE. After resuspension, the plate was read on the Luminex MagPix instrument, where a median fluorescence intensity (MFI) value was generated for each sample and individual antigen with the corresponding bead count. Once the

control MFIs (positive and negative controls), standards (dilution of positive control), and bead counts (minimum 50 beads per antigen in each well) were validated, BSA was subtracted (including for the standards) to account for non-specific binding. Next, we applied an across-plate normalization factor based on the standards and transformed the MFIs into qualitative (positive and negative outcomes) as previously described [22–25]. Previously collected de-identified banked blood/serum samples served as negative (n = 36) and positive (n = 21) controls [8].

## Statistics

Statistical calculations and graphs were completed in R Studio (version 2024.09.0 + 375), GraphPad Prism (version 10.5.0), and the open-source SeroNIST tool (available at the following link: https://github.com/SeroNISTPI/SeroNIST).

Frequency and percentages were used to describe the sociodemographic descriptors and vaccine uptake among the study participants. The Pearson's Chi-squared test with Yates' continuity correction was used to assess differences in vaccine up-take based on sociodemographic variables and by region. The multivariable analysis/logistic regression for COVID-19 vaccine uptake and serology (positive or negative for anti-N SARS-CoV-2 antibodies) were run in R Studio with the glm function [26]. For the vaccine uptake analysis, the missing data points for vaccine uptake (n = 6, 0.8%) were excluded since a binary logistic regression requires an outcome/dependent variable with binary outcomes. For the serological analysis, the missing data points for vaccine uptake (n = 6, 0.8%) were included. The following variables also had missing data: sex (n = 100, 13.4%) and education (n = 54, 7.2%), which were included as separate categories in all the regression analyses. Age had a single missing data point (n = 1), which was not included in any of the models due to the small sample size. The resulting estimates were used to calculate adjusted Odds Ratios (aOR) and 95% confidence intervals (95% CI) which are all listed in the Supplemental Tables.

To visualize anti-SARS-CoV-2 WT and variant antibodies among vaccinated and unvaccinated study participants, raw median fluorescence intensity (MFI) and respective geometric mean and 95% CI were plotted. The geometric mean (vs. standard average) was applied to minimize the effect of outliers. Differences between vaccinated and unvaccinated groups for each SARS-CoV-2 WT and variants were assessed with the unpaired nonparametric Mann–Whitney U test (a.k.a. Wilcoxon rank-sum test). Note, while the graph in Fig 2 is in Log(10) scale to visualize the MFI spread, the statistical test was applied to raw/non-transformed numbers.

We utilized two methods to calculate the seroprevalence of SARS-CoV-2 WT and variants: (1) The first approach was a traditionally applied method where the mean and standard deviation (STDV) of the negative controls is calculated and then the STDV multiplied by 3 and added to the mean (mean+3*STDV = 3STDV threshold) and set as cut off. This method assumes the reads of the negative controls are normally distributed. Since the MFIs of our negative controls were skewed (i.e., not normally distributed data) we attempted to log transformed as previously described [22]. However, log(10) transforming the MFI values of the negative controls for each antigen/isotype skewed the distribution even more, so the raw values were used to calculate the 3STDV, likely underlining the inadequacy of this method for other data sets since negative controls will tend to accrue around zero or the lower limit of detection of the applied assay. Note, negative MFI values (artifact of the assay and not meaningful outcomes since there cannot be a negative value of antibodies in the samples) were set to 1. Similarly, values of zero (n = 6) were set to 1 to be able to show the spread of the data in log(10). See S1 Fig for the raw MFI of the negative and positive controls, along with the calculated 3STDV thresholds. (2) The second approach is a standardized open-source machine learning-based method that calculates 1-, 2-, and 3-dimensional classification boundaries based on confirmed positive and negative controls, as previously described (i.e., SeroNIST) [20–23]. Each dimension corresponds to the number of antigens included, (i.e., equivalent to 1, 2, or 3 antigens). For example, 3 antigens covered WT RBD, N, and S to estimate SARS-CoV-2 seroprevalence, see S2 Fig. The resulting classification boundary is used as threshold, same as for the 3STDV method. For the SeroNIST method, raw MFI values were used (no adjustment for negative values or zeros necessary). The serological platform was evaluated, validated, and used for SARS-CoV-2 seroprevalence studies prior to this, see following references [23–25].

## Results

### Study population and demographics

A total of 745 participants were recruited from Owo in Ondo State (n = 375) from October to November 2022 and from Ikorodu in Lagos State (n = 370) from November 2022 to February 2023. Over half of the participants were female (n = 430, 57.7%), while 28.9% (n = 215) were male, and 13.4% (n = 100) did not report sex. Ages ranged from 0 to 84 years (mean: 35.1 years, STDV: 14.7 years), while one person did not report age (0.1%). The majority of participants had a secondary education (n = 389, 52.2%), followed by those with university education (n = 164, 22.0%), and then those with primary school education (n = 130, 17.4%). A total of 54 individuals (7.2%) did not report education. Christianity was the dominant self-reported religion among our participants (n = 570, 76.5%), followed by Islam (n = 138, 18.5%), and those who practised other religions (n = 36, 4.8%). Only one person reported not practicing any religion (0.1%). These sociodemographic variables are also described in the Supplemental Tables (S1 Table through S5 Table).

### Vaccine uptake

The majority of participants reported having received either a single or double dose of the COVID-19 vaccine (n = 492, 66.0%), while 247 (33.2%) reported not being vaccinated, and six participants (0.8%) opted out of reporting COVID-19 vaccine uptake, see S6 Table. The majority of the participants from Owo (peri-urban, Ondo State) were vaccinated (n = 300, 80.2%, S7 Table). The vaccine uptake in Ikorodu (urban, Lagos State) was also relatively high (n = 192, 52.6%) but lower than in Owo. The difference in regional vaccine uptake was statistically significant (Pearson's Chi-squared test with Yates' continuity correction), where peri-urban study participants from Owo were more likely to be vaccinated than the urban participants from Ikorodu (Chi-square statistic = 62.1, df = 1, p-value < 0.001). To determine whether sex, age, education, religion, state of residency, or being positive for anti-N SARS-CoV-2 antibodies independently increased or decreased the odds of having received a COVID-19 vaccine, we ran a multivariable analysis (i.e., logistic regression). To converge the model, religion and education categories with very small sample sizes (n = 1 for no religion and n = 8 for no formal education) were collapsed into larger categories for this analysis, see S8 and S9 Tables. Residency (Owo), age, and being positive for anti-N WT SARS-CoV-2 antibodies were all independently statistically significant and thus associated with vaccination status, see Fig 1 and S10 Table. Hence, individuals from Owo (aOR = 3.9, p-value < 0.0001) and older study participants (aOR = 1.1, p-value < 0.0001) had significantly higher odds of being vaccinated. Participants positive for anti-N SARS-CoV-2 antibodies (aOR = 0.5, p-value < 0.002) were much less likely to be vaccinated than participants negative for anti-N WT SARS-CoV-2 antibodies. We reran the analysis including age as a categorial variable to confirm these outcomes, see S11 Table. Residency (Owo), increased age, and being positive for anti-N WT SARS-CoV-2 antibodies were statistically significant again, see S12 Table. Individuals who were missing sex descriptors were also statistically significant, but the effect was barely significant (p-value = 0.043) and only observed when age was included as a categorical variable. Thus, in summary, residents from Owo and study participants of older age had much higher odds of being vaccinated than residents from Ikorodu and study participants below the age 15. Further, individuals who had anti-N SARS-CoV-2 antibodies (reflecting natural infection) had lower odds of being vaccinated than those who were negative for anti-N SARS-CoV-2 antibodies. Finally, sex, education, and religion did not influence vaccine uptake in our analysis.

Comparing the presence of anti-RBD SARS-CoV-2 antibodies between those who reported being vaccinated and those who were not vaccinated revealed that vaccinated participants had significantly more anti-RBD antibodies against the WT, Beta, Gamma, Lambda, and Omicron variants (Fig 2). The most significant difference between the vaccinated and unvaccinated group was for RBD Beta (p-value < 0.0001).

### SARS-CoV-2 seroprevalence among unvaccinated study participants

Next, we determined the SARS-CoV-2 seroprevalence among unvaccinated study participants using two different methods. For this purpose, we compared two different strategies to calculate thresholds and determine which samples are

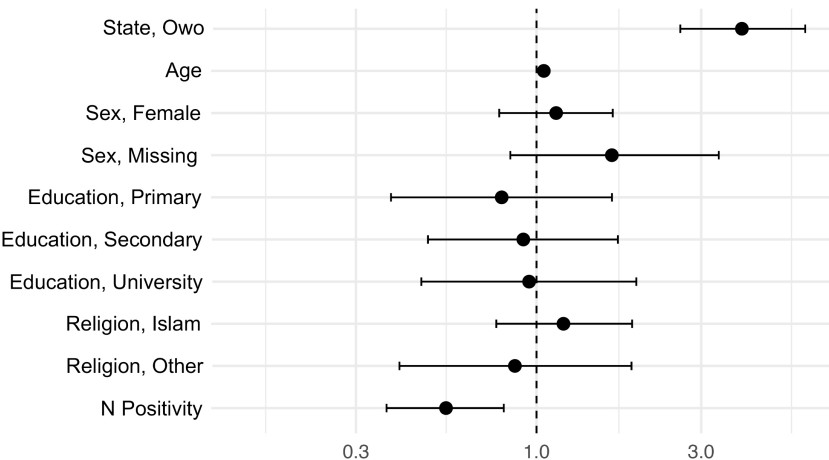

**Fig 1. Adjusted odds ratios for predictors of COVID-19 vaccine uptake.** The figure shows adjusted odds ratios (aORs) for vaccine uptake in log scale by evaluated predictors (state, age, sex, education, religion, and anti-N SARS-CoV-2 seropositivity). Owo residency and age were strongly associated with higher odds of vaccination (p-value < 0.0001), while seropositivity had lower odds of vaccination (p-value = 0.002). Sex, education, and religion were not significantly associated with vaccine uptake.

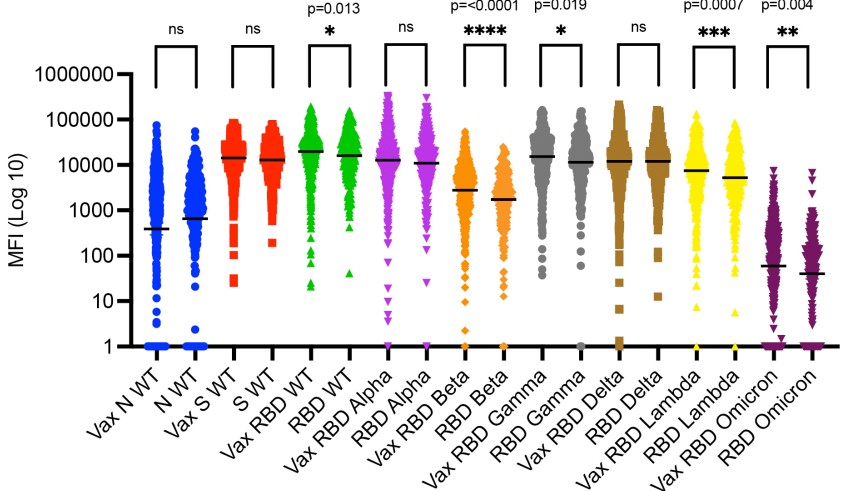

**Fig 2. Anti-SARS-CoV-2 variant antibodies among vaccinated and unvaccinated study participants.** The graph shows raw median fluorescence intensity (MFI) outcomes by variant and the respective geometric means and 95% confidence intervals (CI). Vaccinated participants had significantly higher antibody levels against the wildtype (WT) RBD (p-value = 0.013), RBD Beta (p-value < 0.0001), RBD Gamma (p-value = 0.019), RBD Lambda (p-value = 0.0007), and RBD Omicron (p-value = 0.004) variants, but not RBD Alpha, RBD Delta, and WT Spike (S). Negative MFI values (artifact of the assay and not meaningful outcomes since there cannot be a negative value of antibodies in the samples) were set to 1 to plot the log(10) scale graph and better visualize the full spread of MFIs. Similarly, the few outcomes that ended up being 0 (n = 3 study samples, Omicron only) were set to 1 for the same reasons. Vax, Vaccinated. MFI, median fluorescence intensity. RBD, receptor binding domain. Ns, not significant.

considered positive or negative according to the calculated cut-offs: (1) The first approach was a traditionally applied method where the threshold is set at three standard deviations above the mean of the negative controls (3STDV). (2) The second approach was based on a standardized open-source machine learning-based method that calculates sensitive and specific classification boundaries based on population-specific pre-validated positive and negative controls and gives individual and group-based uncertainty estimates for the seroprevalence outcomes (i.e., SeroNIST) [22–25]. We focused

on calculating SARS-CoV-2 seroprevalence among the unvaccinated study population because we aimed to determine the proportion of participants who had been infected by SARS-CoV-2 (i.e., harbour infection-induced antibodies and not vaccine-induced antibodies).

Overall, the SARS-CoV-2 seroprevalence for anti-S and anti-RBD antibodies across all variants, except Omicron, was high (ranging from 89.1% to 100.0%, Table 1) and similar comparing the two methods among the unvaccinated study population. The seroprevalence for anti-RBD antibodies for the Omicron variant resulted in the most significant discrepancy and resulted in 82.2% (+/- 10.2 95% CI) for the SeroNIST method and 10.5% for the 3STDV method.

To determine whether sex, age, education, religion, city of residence, or vaccination status independently increased or decreased the odds of being positive for anti-N WT SARS-CoV-2 antibodies, we ran a multivariable analysis (i.e., logistic regression with binary seropositivity outcomes, see S8, S9, and S13 Tables). We focused on the N-based seroprevalence because all the COVID-19 vaccines distributed in Nigeria covered S/RBD but not N antigens. Thus, the presence of anti-N SARS-CoV-2 antibodies were predictive of natural SARS-CoV-2 infection and not vaccination in this setting. Further, the RBD- and S-based seroprevalences were very high (ranged from 82.2% to 100%) and preclude a meaningful analysis (i.e., not enough negative outcomes, see Table 1).

When running the multivariable analysis with age as continuous variable, being from Owo (aOR = 1.5, p-value = 0.045), being female (aOR = 1.4, p-value = 0.049), and being vaccinated (aOR = 0.5, p-value = 0.001) were independently statistically significant and were therefore associated with N WT SARS-CoV-2 serology, see Fig 3 and S14 Table. Thus, residents from Owo and female study participants had higher odds of being anti-N WT SARS-CoV-2 antibody positive, though those effects were barely statistically significant. Vaccinated individuals were much less likely to be N WT SARS-CoV-2 antibody positive (i.e., of being infected with SARS-CoV-2), and that effect was strong. There was also a strong association between N WT SARS-CoV-2 seropositivity and not reporting sex (aOR = 7.0, p-value < 0.0001). We reran the analysis including age as a categorial variable to confirm these outcomes. Residency (Owo), being female, being vaccinated, and missing a sex descriptor were statistically significant again, see S15 Table. Thus, in summary, vaccinated individuals had much lower odds of being SARS-CoV-2 positive (likely indicating a protective effect of the vaccine), while there was a mild association between SARS-CoV-2 seropositivity and living in Owo or being female. Age, education, and religion did not influence serological positivity in our analysis. Finally, those who did not report sex were much more likely to be positive for anti-N WT SARS-CoV-2 antibodies, indicating a non-random pattern in missing data.

**Table 1. SARS-CoV-2 Seroprevalence among unvaccinated participants (n = 247) by threshold assessment method (3STDV vs. SeroNIST).**

| | 3STDV | SeroNIST | | |
|---|---|---|---|---|
| | Seroprevalence % | Seroprevalence % (95% CI) | Sensitivity (95% CI) | Specificity (95% CI) |
| N WT | 66.4 | 78.9 (+/- 7.7) | 100.0 (98.8, 100.0) | 72.2 (59.6, 84.2) |
| S WT | 91.9 | 100.0 (+0/-3.4) | 100.0 (98.8, 100.0) | NA |
| RBD WT | 95.1 | 100.0 (+0/-8.9) | 100.0 (98.3, 100.0) | NA |
| RBD Alpha | 89.1 | 100.0 (+0/-7.0) | 100.0 (98.8, 100.0) | NA |
| RBD Beta | 96.0 | 100.0 (+0/-3.4) | 100.0 (98.8, 100.0) | NA |
| RBD Gamma | 97.2 | 100.0 (+0/-5.6) | 100.0 (98.8, 100.0) | NA |
| RBD Delta | 99.6 | 99.6 (+/- 3.2) | 100.0 (98.8, 100.0) | NA |
| RBD Lambda | 98.0 | 100.0 (+0/-2.8) | 100.0 (98.8, 100.0) | NA |
| RBD Omicron | 10.5 | 82.2 (+/- 10.2) | 100.0 (98.8, 100.0) | 80.6 (68.2, 91.5) |

NA, not applicable because there were too few negative outcomes to evaluate specificity for this analysis.

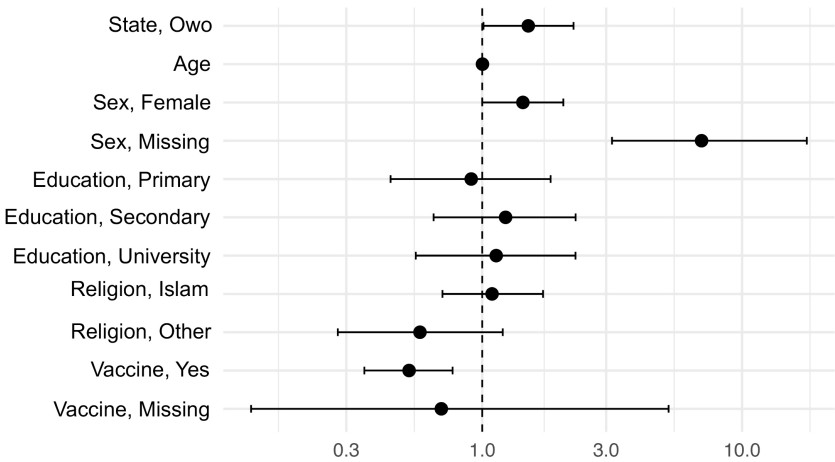

**Fig 3. Adjusted odds ratios for predictors of anti-N SARS-CoV-2 seropositivity.** The figure shows adjusted odds ratios (aORs) for anti-N wildtype (WT) SARS-CoV-2 seropositivity by evaluated predictors (state, age, sex, education, religion, and COVID-19 vaccine uptake) in form of a forest plot. Vaccinated individuals had much lower odds of being positive (p-value = 0.001). Owo residency (p-value = 0.045) and being female (p-value = 0.049) were associated with higher odds of vaccination, though the effects were barely significant. Those who did not report sex were much more likely to be positive for anti-N WT SARS-CoV-2 antibodies (p-value < 0.0001). Age, education, and religion did not influence serological positivity. Age, education, and religion did not influence serological positivity.

## Discussion

The COVID-19 pandemic, caused by SARS-CoV-2, rapidly spread across the globe in 2019/2020, resulting in millions of deaths, overwhelming healthcare systems, disrupting economies, and exposing vulnerabilities in global public health [27,28]. Surprisingly, the number of cases and fatalities in Africa was low compared to the rest of the world [29]. Nigeria, Africa's most populous country, accounted for less than 1% of the global cases [2]. Whether these numbers reflect the true burden of SARS-CoV-2 is yet to be elucidated. The present study evaluated serosurvey-based SARS-CoV-2 exposure, compared serological threshold assessment approaches, determined the risk factors for SARS-CoV-2 infection, and evaluated the COVID-19 vaccine uptake among febrile Southern Nigerian study participants across urban and peri-urban settings in 2022/2023. These study results offer insights into sociodemographic and regional risk factors for SARS-CoV-2 infection, strategies to optimize serological study design, and approaches to improve COVID-19 vaccine uptake in West Africa.

### Study population

We found slightly higher female involvement and most of the participants to be between 25 and 44 years old, which aligns with previous studies showing greater engagement in public health research by women and socially active populations [30,31]. This was also reflected in the educational status of the participants, where most people had received post-primary formal education.

### Vaccine uptake

The majority of study participants (66.0%) reported receiving a single or double dose of the COVID-19 vaccine, which is above the previously reported COVID-19 vaccine uptake in Nigeria, ranging from 20%-58% [32]. While prior research indicates that sociodemographic factors significantly influence vaccine uptake [33–36], we did not find sex, education, or religion to be strong indicators for COVID-19 vaccine uptake in our study population. However, we found that peri-urban

study participants from Owo were more likely to be vaccinated compared to urban counterparts from Ikorodu, indicating differential COVID-19 vaccine uptake across urban gradients in our study area. Since basic socioeconomic factors were not significant determinants of vaccine uptake in our study population, it may have been due to variation in COVID-19 vaccine distribution and accessibility or other factors our study did not capture. Furthermore, we found that older age was strongly associated with the likelihood of being vaccinated, likely reflecting the local vaccine campaign efforts during the study period, which focused on 60+ adults and 50+ adults with co-morbidities [37].

Interestingly, study participants who were anti-N SARS-CoV-2 antibody positive (reflecting natural infection) had much lower odds of being vaccinated. The statistically significant relationship between vaccine uptake and anti-N SARS-CoV-2 antibody positivity was confirmed through subsequent analyses (serology as dependent variable), where vaccinated individuals were much less likely to be N SARS-CoV-2 antibody positive. The effect in both models was strong and may indicate a protective effect of the vaccine, supported by finding higher anti-RBD WT antibodies among vaccinated study participants. The effect could also reflect or be compounded by behavioural modifications where cautious individuals were both more likely to get vaccinated and better at social distancing and hand-washing protocols, therefore preventing SARS-CoV-2 infection during the study period.

## SARS-CoV-2 serosurvey

Vaccinated participants had significantly more anti-RBD antibodies for the WT, Beta, Gamma, Lambda, and Omicron variants (Fig 2). Detecting more anti-S/RBD antibodies in the vaccinated group likely validated the self-reported vaccination status since (i) being vaccinated has been described to elicit a stronger anti-RBD/S antibody response than natural infection, and (ii) being both vaccinated and infected is likely to induce a higher antibody response than either alone [8,38]. However, antibody decay should be taken into account [39,40], and we did not have information on time since vaccination or infection.

## SARS-CoV-2 serosurvey among unvaccinated study participants

We assessed SARS-CoV-2 seroprevalence among unvaccinated study participants to evaluate SARS-CoV-2 exposure (vs. vaccination). Our study revealed high S/RBD-based seroprevalences for all variants except Omicron among the unvaccinated participants (ranging from 89.1% to 100%), indicating that the level of SARS-CoV-2 exposure was likely underestimated in Nigeria and confirming that reduced viral transmission was not likely the main reason the reported COVID-19 morbidity and mortality were low in Nigeria and the region [6]. This is the highest seroprevalence rate for SARS-CoV-2 reported in Nigeria to date [11–16]. The variation may reflect differences in sampling period and testing approach, advocating standardized serological methods. While most of the prior serosurveys were implemented between 2020 and mid-2022, our study was conducted later in the pandemic, between the end of 2022 and beginning of 2023. A similar high seroprevalence of over 90% was reported in Yemen in 2023 [41].

Furthermore, Nigerians may have been exposed to SARS-CoV-2 variants that were not previously reported in the country. Nigeria experienced four major SARS-CoV-2 waves dominated by the ancestral/WT, Alpha, Delta, Eta, and Omicron variants [17–20]. Other important variants detected elsewhere, such as Gamma and Lambda variants, were not reported in Nigeria [42], but we detected relatively high levels of anti-RBD antibody for both variants among our study population, although some of the signal may have been due to cross-reactivity. While the affinity between the specific variant antigens (i.e., antigens coupled to the beads in our serological assay) and respective human antibodies should be the highest/ tightest, we could not preclude cross-reactivity between SARS-CoV-2 variants and also with endemic human CoVs (e.g., HKU1, NL63, OC43, and 229E) due to lack of positive controls (i.e., individuals who were exposed only to specific SARS-CoV-2 variants, HKU1, NL63, OC43, or 229E). The low seroprevalence and antibody levels for Omicron RBD detected in our study may have been due to low transmission or due to a distinct serological footprint [43]. Further genomic epidemiological approaches and proactive serosurveillance will help unravel the true burden of all SARS-CoV-2 variants in Nigeria.

## Risk factors for SARS-CoV-2 infection

We evaluated risk factors for SARS-CoV-2 infection through an anti-N WT SARS-CoV-2 serosurvey-based multivariable analysis across all study participants (vaccinated and unvaccinated). Since vaccines distributed in Nigeria during the study period covered S/RBD but not N antigens, we focused on the N-based seroprevalence to estimate risk factors for natural SARS-CoV-2 infection (vs. vaccination). Further, the RBD- and S-based seroprevalences were high and ranged from 82.2% to 100.0%, precluding a meaningful risk factor analysis.

As mentioned above, vaccinated individuals were much less likely to be N WT SARS-CoV-2 antibody positive (p-value = 0.001). Female study participants had higher odds of being positive for anti-N WT SARS-CoV-2 antibodies, though the effect was barely statistically significant (p-value = 0.049). Age, education, and religion did not influence the serological outcome. Thus, we did not find compelling high-risk sociodemographic subpopulations based on age, sex, educational status, or religion, similar to previous studies [11,30]. Finally, Owo residents were more likely to be N WT SARS-CoV-2 positive. Though the effect was barely statistically significant (p-value = 0.045), study participants living in peri-urban settings were at higher risk of SARS-CoV-2 infection compared to urban settings (Ikorodu).

## Serological threshold assessment

Our study compared two threshold calculation methods to evaluate the SARS-CoV-2 seroprevalence. There was concordance between the two approaches, except for anti-RBD Omicron and anti-N WT antibodies. The difference between the N-based seroprevalence was within ~10% when considering the 95%CI, but the Omicron-based seroprevalence differed widely (82.2% vs. 10.5%). The SeroNIST method calculates thresholds by utilizing measurements from positive and negative controls (whereas the 3STDV approach is based on only the negative controls) and the positive controls were cases captured before Omicron circulated in the local population. Hence, they were not good positive controls to determine the SeroNIST-based thresholds for Omicron. Overall, the two methods estimated similar seroprevalence outcomes (except for Omicron, where no appropriate positive controls were available). However, the SeroNIST method represents a more rigorous and standardized approach. It automatically calculates seroprevalence and the associated uncertainty estimates (sensitivity, specificity, and accuracy) for the population-based and individual serological outcomes. Using an open source, standardized, and automated approach across serosurveys (independent of the pathogen of interest) will make seroprevalence calculations quicker, more informative, and comparable across studies, making the outcomes more suitable for metanalysis and increasing the public health impact of serosurveys overall. Hence, we considered the SeroNIST method superior to the 3SDTV approach.

## Limitations

Our study results should be viewed considering a few limitations. While we were able to enroll 745 study participants, increasing the limited sample size, including additional socioeconomic variables, and being able to access vaccination dates would have given a more nuanced picture of risk factors for SARS-CoV-2 exposure and COVID-19 vaccine uptake. Unfortunately, that information was not available to us. Further, the socio-demographic information and vaccine uptake were self-reported which may lead to recall bias or misrepresentations.

We had missing data for vaccine uptake (n = 6, 0.8%), sex (n = 100, 13.4%), education (n = 54, 7.2%), and age (n = 1/275). All the missing data was included as separate categories in the regression analyses, except for age. There were no strong and significant outcomes for the missing data categories except for sex. Those who did not report sex were much more likely to be positive for anti-N SARS-CoV-2 antibodies, indicating a non-random pattern in missing data. However, anti-N SARS-CoV-2 antibody positivity was not a self-reported outcome and therefore not likely to be directly related.

Importantly, our study only covered symptomatic participants who visited local healthcare centers. Hence, it does not represent the general population. Further, individuals who visit health care centers when sick may be more open

to engage with the public health system and therefore may be more likely to get vaccinated compared to the general population.

Finally, the main methodological limitation of our study was the inability to account for potential serological cross-reactivity between SARS-CoV-2 variants and potentially other CoVs due to the lack of positive controls for each CoV subtype.

## Conclusion and recommendations

(i) The results of this study suggest that the SARS-CoV-2 exposure in Southern Nigeria was higher than previously reported. Hence, lack of SARS-CoV-2 transmission was not likely the reason that Nigeria, and potentially the region, experienced less COVID-19 disease and related deaths. Still, genomic surveillance is needed to track variance-based exposure patterns [44]. (ii) The COVID-19 vaccine uptake was high, with peri-urban participants showing greater coverage than those in urban areas, and increasing age was a strong predictor of vaccination. Hence, the local vaccine uptake strategy implemented prior to 2022/2023 proved effective, resulting in high coverage rates and representing a significant public health success. Interestingly, our study found a strong correlation between being vaccinated and having lower odds of being positive for anti-N WT SARS-CoV-2 antibodies (proxy for SARS-CoV-2 infection). Therefore, the vaccine may have protected study participants from SARS-CoV-2 infection prior to the study. Overall, ongoing public health interventions should focus on improving vaccine uptake in urban populations and younger age groups while leveraging the observed success of peri-urban vaccine campaigns. (iii) The lack of standardized serological thresholding methods highlights the need for robust analytical approaches. While there was concordance between the two approaches, prioritizing the use of validated, standardized, and automated thresholding methods such as SeroNIST will enable reliable comparisons of serosurveys across studies and time, thereby increasing the public health impact of serosurveys.

## Supporting information

**S1 Fig. Controls and 3STDV threshold.** The graph shows raw median fluorescence intensity (MFI) outcomes of the positive controls (red dots) and the negative controls (blue dots) by SARS-CoV-2 variant. In green are the 3STDV thresholds (based on negative controls only), with respective values on top. Note, negative MFI values (artifact of the assay and not meaningful outcomes since there cannot be a negative value of antibodies in the samples) were set to 1 to plot the log(10) scale graph and better visualize the full spread of MFIs. MFI, median fluorescence intensity. S, Spike. RBD, receptor binding domain. N, Nucleoprotein.
(PDF)

**S2 Fig. SeroNIST classification boundary based on three antibodies.** (A) This figure shows the classification boundary (i.e., three-dimensional threshold in rainbow colours) created by the SeroNIST fool for wildtype (WT) RBD, S, and N antibodies based on the training data (i.e., negative control in blue dots and positive controls in red dots). (B) This figure shows where the data points (blue dots) fall for an example data set of participants. In this case, all the data points fall above the classification boundary, resulting in an estimated SARS-CoV-2 seroprevalence of ~100% based on RBD, S, and N antibodies.
(PDF)

**S1 Table. Residency of study participants (n = 745).**
(XLSX)

**S2 Table. Sex distribution among study participants (n = 745).**
(XLSX)

**S3 Table. Age of study participants (by category, n = 745).**
(XLSX)

**S4 Table. Educational level of study participants (n = 745).**
(XLSX)

**S5 Table. Religion among study participants (n = 745).**
(XLSX)

**S6 Table. Vaccine uptake among study participants (n = 745).**
(XLSX)

**S7 Table. Vaccine uptake by region/state.**
(XLSX)

**S8 Table. Education variable for multivariable analysis.** No formal education (n = 8) and primay education (n = 130) were combined (n = 138).
(XLSX)

**S9 Table. Religion variable for multivariable analysis.** No religion (n = 1) and Other (n = 36) were combined (n = 37).
(XLSX)

**S10 Table. Model outputs for vaccine uptake multivariable analysis (age as continuous variable).**
(XLSX)

**S11 Table. Categorical age variable for vaccine uptake multivariable analysis.**
(XLSX)

**S12 Table. Model outputs for vaccine uptake multivariable analysis (age as categorical variable).**
(XLSX)

**S13 Table. N WT SARS-CoV-2 seropositivity among study participants.**
(XLSX)

**S14 Table. Model outputs for N seropositivity multivariable analysis (age as continuous variable).**
(XLSX)

**S15 Table. Model outputs for N seropositivity multivariable analysis (age as categorical variable).**
(XLSX)

**S1 Checklist. The checklist contains information around ethical concerns of global research.**
(DOCX)

**S1 Data. This supplemental database contains the data used for the performed analyses.**
(XLSX)

## Acknowledgments

We thank the study participants and their families for their valuable time and participation. Without them, this study would not have been possible.

## Author contributions

**Conceptualization:** Ann M. Moormann, Rhoel R. Dinglasan, Christian T. Happi, Raquel A. Binder.

**Data curation:** Chinedu A. Ugwu, Ooreofe Odebode, Israel O. Ajayi, Josiah Olubunmi, Bejide Ifeoluwa, Jolly A. Adole, Kazeem Akano, Johnson Okolie, Philomena Eromon, Imonikhe K. Kio, Judith Amadi, Precious F. Adebayo, Iheanyi

Mgbeokwere, Akeem O. Lawal, Olayemisi A. Adegoke, Amoke O. Adeyemi, Sampson Owhin, Oladele O. Ayodeji, Alebiosu Ahmed, Etim Henshaw, Oyejide Nicholas Eyitayo, Abah A. Sylvester, Abiola Omidele, Kolade Emilola, Olayiwola Funmilayo, Adeboye Oyewale Rufus, Oluwatuyi O. Olakunle, Obaado Oluwaseun, Prajakta Bedekar, Latifeh Dahmash, Catherine S. Forconi.

**Formal analysis:** Chinedu A. Ugwu, Angela M. Matta, Prajakta Bedekar, Catherine S. Forconi, Raquel A. Binder.

**Funding acquisition:** Ann M. Moormann, Rhoel R. Dinglasan, Christian T. Happi, Raquel A. Binder.

**Investigation:** Chinedu A. Ugwu, Ooreofe Odebode, Israel O. Ajayi, Josiah Olubunmi, Bejide Ifeoluwa, Jolly A. Adole, Kazeem Akano, Johnson Okolie, Philomena Eromon, Imonikhe K. Kio, Judith Amadi, Precious F. Adebayo, Iheanyi Mgbeokwere, Akeem O. Lawal, Olayemisi A. Adegoke, Amoke O. Adeyemi, Sampson Owhin, Oladele O. Ayodeji, Alebiosu Ahmed, Etim Henshaw, Oyejide Nicholas Eyitayo, Abah A. Sylvester, Abiola Omidele, Kolade Emilola, Olayiwola Funmilayo, Adeboye Oyewale Rufus, Oluwatuyi O. Olakunle, Obaado Oluwaseun, Angela M. Matta, Prajakta Bedekar, Latifeh Dahmash, Catherine S. Forconi, Ann M. Moormann, Rhoel R. Dinglasan, Christian T. Happi, Raquel A. Binder.

**Methodology:** Prajakta Bedekar, Latifeh Dahmash, Catherine S. Forconi, Raquel A. Binder.

**Project administration:** Chinedu A. Ugwu, Ooreofe Odebode, Israel O. Ajayi, Josiah Olubunmi, Bejide Ifeoluwa, Jolly A. Adole, Kazeem Akano, Johnson Okolie, Philomena Eromon, Imonikhe K. Kio, Judith Amadi, Precious F. Adebayo, Iheanyi Mgbeokwere, Akeem O. Lawal, Olayemisi A. Adegoke, Amoke O. Adeyemi, Sampson Owhin, Oladele O. Ayodeji, Alebiosu Ahmed, Etim Henshaw, Oyejide Nicholas Eyitayo, Abah A. Sylvester, Abiola Omidele, Kolade Emilola, Olayiwola Funmilayo, Adeboye Oyewale Rufus, Oluwatuyi O. Olakunle, Obaado Oluwaseun, Ann M. Moormann, Rhoel R. Dinglasan, Christian T. Happi.

**Resources:** Ann M. Moormann, Rhoel R. Dinglasan, Christian T. Happi.

**Supervision:** Chinedu A. Ugwu, Latifeh Dahmash, Ann M. Moormann, Rhoel R. Dinglasan, Christian T. Happi.

**Validation:** Raquel A. Binder.

**Visualization:** Chinedu A. Ugwu, Raquel A. Binder.

**Writing – original draft:** Chinedu A. Ugwu, Raquel A. Binder.

**Writing – review & editing:** Chinedu A. Ugwu, Ooreofe Odebode, Israel O. Ajayi, Josiah Olubunmi, Bejide Ifeoluwa, Jolly A. Adole, Kazeem Akano, Johnson Okolie, Philomena Eromon, Imonikhe K. Kio, Judith Amadi, Precious F. Adebayo, Iheanyi Mgbeokwere, Akeem O. Lawal, Olayemisi A. Adegoke, Amoke O. Adeyemi, Sampson Owhin, Oladele O. Ayodeji, Alebiosu Ahmed, Etim Henshaw, Oyejide Nicholas Eyitayo, Abah A. Sylvester, Abiola Omidele, Kolade Emilola, Olayiwola Funmilayo, Adeboye Oyewale Rufus, Oluwatuyi O. Olakunle, Obaado Oluwaseun, Angela M. Matta, Prajakta Bedekar, Latifeh Dahmash, Catherine S. Forconi, Ann M. Moormann, Rhoel R. Dinglasan, Christian T. Happi, Raquel A. Binder.

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
