## [Decision Letter · Decision Letter 0]

11 Nov 2025

PGPH-D-25-02354

COVID19 Sero-epidemiology and Vaccine Uptake in Two Southern Nigerian Clinics

Dear Dr. Binder,

Thank you for submitting your manuscript to PLOS Global Public Health. After careful consideration, we feel that it has merit but does not fully meet PLOS Global Public Health’s publication criteria as it currently stands. Therefore, we invite you to submit a revised version of the manuscript that addresses the points raised during the review process.

We look forward to receiving your revised manuscript.

Kind regards,

Kate Zinszer

Academic Editor

Journal Requirements:

2. Please provide a detailed online Financial Disclosure statement. This is published with the article. It must therefore be completed in full sentences and contain the exact wording you wish to be published.

a) State the initials, alongside each funding source, of each author to receive each grant. For example: “This work was supported by the National Institutes of Health (####### to AM; ###### to CJ) and the National Science Foundation (###### to AM).”

For more information, please go to our submission guidelines:

https://journals.plos.org/globalpublichealth/s/submission-guidelines#loc-financial-disclosure-statement

3. Please ensure that the funders and grant numbers match between the Financial Disclosure field and the Funding Information tab in your submission form. Note that the funders must be provided in the same order in both places as well.

4. We note that your Data Availability Statement is currently as follows: “The full database has been submitted along with the manuscript and figures.”

Please confirm at this time whether or not your submission contains all raw data required to replicate the results of your study. Authors must share the “minimal data set” for their submission. PLOS defines the minimal data set to consist of the data required to replicate all study findings reported in the article, as well as related metadata and methods (https://journals.plos.org/globalpublichealth/s/data-availability#loc-minimal-data-set-definition).

If your submission does not contain these data, please either upload them as Supporting Information files or deposit them to a stable, public repository and provide us with the relevant URLs, DOIs, or accession numbers. For a list of recommended repositories, please see https://journals.plos.org/globalpublichealth/s/recommended-repositories.

5. Please provide separate main figure files in .tif or .eps format only and ensure that all files are under our size limit of 10MB.

Additional Editor Comments (if provided):

Please address the comments brought forth by the reviewers with a focus on including a more comprehensive multivariable analysis for factors associated with seroprevalence in the total study population and in the unvaccinated study population. The potential confounders included in the model should be justified based on literature and 95% CIs should be included in tables and in the text when referencing results from the regression analyses. Clear tables should be included in the manuscript that identify the results from the regression model(s). Also, stronger justification for excluding missing data should be included vs imputation. Where the patterns of missingness examined?

Reviewers' comments:

Reviewer's Responses to Questions

**Comments to the Author**

1. Does this manuscript meet PLOS Global Public Health’s publication criteria ? Is the manuscript technically sound, and do the data support the conclusions? The manuscript must describe methodologically and ethically rigorous research with conclusions that are appropriately drawn based on the data presented.

Reviewer #1: Yes

Reviewer #2: Yes

2. Has the statistical analysis been performed appropriately and rigorously?

Reviewer #1: Yes

Reviewer #2: Yes

3. Have the authors made all data underlying the findings in their manuscript fully available (please refer to the Data Availability Statement at the start of the manuscript PDF file)?

Reviewer #1: Yes

Reviewer #2: Yes

4. Is the manuscript presented in an intelligible fashion and written in standard English?

Reviewer #1: Yes

Reviewer #2: Yes

Reviewer #1: Dear Sr/Madam

I have now completed my review of the paper and incorporated my comments directly into the text for your consideration. The main recommendations focus on:

1. Inclusion criteria considering that sampling is limited to febrile outpatients, and so generalizability is reduced.

2. The fact that vaccination data is self-reported. You may want to consider clarifying the risk of misclassification.

3. On the statistics, there is limited multivariate analysis. I recommend stronger modelling for uptake predictors.

4. No explicit herd immunity threshold or practical cold-chain/coverage was discussed. If possible, it should be reviewed.

Please let me know if any of the changes require further clarification or if you would like to discuss any of the points raised.

Kind regards,

Maria Chimpolo

Reviewer #2: The study COVID-19 Sero-epidemiology and Vaccine Uptake in Two Southern Nigerian Clinics is certainly relevant and important. However, my main comment: Please ensure that the statistics reported in the results narrative correspond exactly to those in the tables. Use totals that exclude missing values (e.g., Table 1) and avoid referencing any statistics not presented in the tables. This consistency is critical for data accuracy.

**Do you want your identity to be public for this peer review?** For information about this choice, including consent withdrawal, please see our Privacy Policy .

Reviewer #1: No

Reviewer #2: **Yes:** Geofrey BasalirwaGeofrey Basalirwa

---

## [Decision Letter · Decision Letter 1]

12 Feb 2026

PGPH-D-25-02354R1

COVID19 Sero-epidemiology and Vaccine Uptake in Two Southern Nigerian Clinics

Dear Dr. Binder,

Thank you for submitting your manuscript to PLOS Global Public Health. After careful consideration, we feel that it has merit but does not fully meet PLOS Global Public Health’s publication criteria as it currently stands. Therefore, we invite you to submit a revised version of the manuscript that addresses the points raised during the review process.

The manuscript has been evaluated by three reviewers, and their comments are available below. Please address the remaining minor comments from Reviewer 2.

We look forward to receiving your revised manuscript.

Kind regards,

Alejandro Torrado Pacheco, PhD

Associate Editor

Journal Requirements:

Additional Editor Comments (if provided):

Reviewers' comments:

Reviewer's Responses to Questions

**Comments to the Author**

Reviewer #1: All comments have been addressed

Reviewer #2: All comments have been addressed

Reviewer #3: All comments have been addressed

publication criteria ? Is the manuscript technically sound, and do the data support the conclusions? The manuscript must describe methodologically and ethically rigorous research with conclusions that are appropriately drawn based on the data presented.

Reviewer #1: Yes

Reviewer #2: Yes

Reviewer #3: Yes

3. Has the statistical analysis been performed appropriately and rigorously?

Reviewer #1: Yes

Reviewer #2: Yes

Reviewer #3: Yes

4. Have the authors made all data underlying the findings in their manuscript fully available (please refer to the Data Availability Statement at the start of the manuscript PDF file)?

Reviewer #1: Yes

Reviewer #2: Yes

Reviewer #3: Yes

5. Is the manuscript presented in an intelligible fashion and written in standard English?

Reviewer #1: Yes

Reviewer #2: Yes

Reviewer #3: Yes

Reviewer #1: I believe that the authors have satisfactorily addressed the substantive scientific and methodological comments from the previous review round. The added multivariable analyses, clearer limitations, and strengthened interpretation make the manuscript suitable for publication. However, minor editorial and consistency issues remain:

1. Results narrative vs tables/figures: In the Results section, some percentages are described without explicitly stating that missing values are excluded, while the corresponding tables use reduced denominators (e.g., comparison between narrative descriptions of seroprevalence and Table 1 totals. Consider aligning text strictly with table denominators. Also, “Age was the strongest predictor…, while sex and education were…”, with the corresponding adjusted odds ratios and confidence intervals could be explicitly linked to a table reference (e.g., “Table 3”) in the narrative, which would strengthen transparency.

2. Terminology consistency: The assay threshold terminology (e.g., variants of the “3 SDTV” threshold notation) is not used consistently across the Methods and Results sections

3. Figure caption wording: Figure 3 caption contains a duplicated sentence and a phrase that appears inconsistent with the figure content (reference to “odds of vaccination” in a seropositivity figure)

4. Comparative numerical results: Please consider presenting results consistently from lowest to highest values when highlighting differences between methods, e.g., reporting the lower estimate first (3STDV: 10.5%) followed by the higher estimate (SeroNIST: 82.2%, 95% CI ±10.2) would make the contrast between methods easier to follow.

These are straightforward corrections that do not affect the validity of the findings. Assuming all underlying datasets and supporting information have been submitted as stated, and subject to these minor revisions, I support publication in PLOS Global Public Health.

Reviewer #2: Review and address these minor attached comments and you will be good to go

Reviewer #3: This manuscript presents a thorough and valuable investigation of SARS-CoV-2 seroprevalence, vaccine uptake, and methodological differences in serological threshold setting among febrile patients affiliated with peri-urban and urban clinics in Southern Nigeria. The comparison between peri-urban and urban patients seems to be a relatively novel consideration. Several vaccines that were in use during the relevant period were also accounted for in the study.

The study yields valuable insights into SARS-Cov-2 seroprevalence, vaccine uptake, and seroprevalence estimates in relation to several demographic factors. Some of these results are interesting and would be unexpected. In particular, the finding that basic socioeconomic factors were not significant determinants of vaccine uptake is of interest.

The proposed framework demonstrates potential applicability to other contexts as well.

The literature review is thorough and the methodology is robust.

Corrections suggested by the original reviewers have been carefully incorporated. These changes have significantly improved the manuscript.

Please include in-text citations using brackets, not inside parentheses.

I have highlighted some words/phrases with comments for potential re-wording for clarity. Please hover over highlighted text (in light blue) to read the comments. Please carefully review that feedback and revise the sentences accordingly.

**Do you want your identity to be public for this peer review?** For information about this choice, including consent withdrawal, please see our Privacy Policy .

Reviewer #1: No

Reviewer #2: **Yes:** Geofrey BasalirwaGeofrey Basalirwa

Reviewer #3: No

---

## [Editor Report · Decision Letter 2]

2 Mar 2026

COVID19 Sero-epidemiology and Vaccine Uptake in Two Southern Nigerian Clinics

PGPH-D-25-02354R2

Dear Dr. Binder,

We are pleased to inform you that your manuscript 'COVID19 Sero-epidemiology and Vaccine Uptake in Two Southern Nigerian Clinics' has been provisionally accepted for publication in PLOS Global Public Health.

Best regards,

Julia Robinson

Executive Editor